# Mushroom-Derived Compounds as Metabolic Modulators in Cancer

**DOI:** 10.3390/molecules28031441

**Published:** 2023-02-02

**Authors:** Bhoomika Dowaraka-Persad, Vidushi Shradha Neergheen

**Affiliations:** 1Biopharmaceutical Unit, Centre for Biomedical and Biomaterials Research (CBBR), University of Mauritius, Réduit 80837, Mauritius; 2Department of Health Sciences, Faculty of Medicine and Health Sciences, University of Mauritius, Réduit 80837, Mauritius

**Keywords:** cancer, conjugated linoleic acid, GL22, grifolin, metabolic reprogramming, neoalbaconol, panepoxydone, (22E, 24R)-6β-methoxyergosta-7,9(11), 22-triene-3β,5α-diol

## Abstract

Cancer is responsible for lifelong disability and decreased quality of life. Cancer-associated changes in metabolism, in particular carbohydrate, lipid, and protein, offer a new paradigm of metabolic hits. Hence, targeting the latter, as well as related cross-linked signalling pathways, can reverse the malignant phenotype of transformed cells. The systemic toxicity and pharmacokinetic limitations of existing drugs prompt the discovery of multi-targeted and safe compounds from natural products. Mushrooms possess biological activities relevant to disease-fighting and to the prevention of cancer. They have a long-standing tradition of use in ethnomedicine and have been included as an adjunct therapy during and after oncological care. Mushroom-derived compounds have also been reported to target the key signature of cancer cells in in vitro and in vivo studies. The identification of metabolic pathways whose inhibition selectively affects cancer cells appears as an interesting approach to halting cell proliferation. For instance, panepoxydone exerted protective mechanisms against breast cancer initiation and progression by suppressing lactate dehydrogenase A expression levels and reinducing lactate dehydrogenase B expression levels. This further led to the accumulation of pyruvate, the activation of the electron transport chain, and increased levels of reactive oxygen species, which eventually triggered mitochondrial apoptosis in the breast cancer cells. Furthermore, the inhibition of hexokinase 2 by neoalbaconol induced selective cytotoxicity against nasopharyngeal carcinoma cell lines, and these effects were also observed in mouse models. Finally, GL22 inhibited hepatic tumour growth by downregulating the mRNA levels of fatty acid-binding proteins and blocking fatty acid transport and impairing cardiolipin biosynthesis. The present review, therefore, will highlight how the metabolites isolated from mushrooms can target potential biomarkers in metabolic reprogramming.

## 1. Introduction

The quest for new small molecules and the repurposing of others is ongoing for disease treatment. It should be noted that 90% of existing therapeutic drugs are small molecules and that the latter provide much hope for the treatment of both communicable and chronic diseases [1].With an estimated number of new cancer cases on the rise and cancer being the top leading cause of death globally, small molecules make an important component of targeted drugs in cancer therapy [2,3]. Several small molecules have been approved by the Federal Drug Administration for cancer treatment, which targets disrupted kinases, epigenetic regulatory proteins, DNA damage repair enzymes, and proteasomes [2].The search for new molecules continues as the existing therapies are unable to contain the burden of the disease. Cancer cells undergo numerous changes in their metabolic pathways, involving energy and biosynthetic processes, so that they can proliferate. Hence, the metabolic pathways appear as interesting targets for a broad spectrum of therapeutic approaches [4]. Mushroom-derived secondary metabolites from edible and medicinal species can affect multiple cancer-related processes. In Japan and China, medicinal mushrooms are approved adjuncts to standard cancer treatment [5]. Hence, this review will investigate the status of mushroom-derived molecules in targeted cancer therapy, with a focus on their mechanisms of action and potential for use.

## 2. Glucose Metabolism and the Warburg Effect

In normal cells, pyruvate produced during glycolysis enters the mitochondria and undergoes an oxidative transformation to form acetyl coenzyme A which then fuels the tricarboxylic acid cycle (TCA), or Krebs cycle, to produce carbon dioxide and water [6]. However, in the absence of oxygen, pyruvate is reduced to lactate in the anaerobic glycolysis pathway, and cancer cells resort to lactic acid fermentation, even in the presence of oxygen. This shift is termed the Warburg effect or aerobic glycolysis [4,7,8,9,10,11]. In fact, neoplasia is the result of this metabolic selection and deregulation of metabolic fluxes constitute an emerging hallmark of cancer [12]. The latter is characterised by the upregulation of glucose transporters, glycolytic enzymes such as hexokinase (HK), phosphofructokinase (PFK), pyruvate kinase (PK), lactate dehydrogenase (LDH), and monocarboxylate transporters [13]. In addition, pathways downstream of oncogenes, such as phosphoinositide 3-kinase (PI3K), protein kinase B (AKT), mammalian target of rapamycin (mTOR), hypoxia-inducible factor (HIF), and c-MYC, and tumour suppressors, such as phosphatase and tensin homolog (PTEN), liver kinase B1, and von Hippel-Lindau, possibly mediated by AMP-activated protein kinase (AMPK) and p53 have been reported [4,14,15]. Recurrent mutations in genes that encode the enzymes aconitase, isocitrate dehydrogenase (IDH), succinate dehydrogenase (SDH), and fumarate hydratase (FH) are also observed in neoplastic cells [16].

Several authors have argued that targeting deranged energy metabolism of tumours, in particular, their hyperbolic addition to glycolysis may provide effective approaches to hinder tumourigenesis [4,7,17,18,19]. This is primarily because this metabolic autonomy emerges early in metastatic colonisation, and the metabolic profile of cancer cells differs significantly from normal cells [6,20]. Furthermore, low oxygen concentration is the first substrate limitation confronting the clonogenicity of transformed cells and short intermittent pulses of hypoxia select cells with a pronounced glycolytic phenotype, and this disordered metabolism also results in the generation of biochemical equivalents in the form of adenosine triphosphate (ATP) to maintain energy homeostasis and electrochemical gradients in cancer cells [9,20,21]. These metabolic dependencies also allow dividing cells to use intermediate glucose metabolites to fuel the pentose phosphate pathway, de novo lipogenesis, the serine biosynthesis pathway, the glycogen shunt, the hexosamine biosynthetic pathway, and de novo glutamine synthesis to double their biomass, and they also confer direct signalling functions to tumour cells to modulate reactive oxygen species (ROS) and chromatin state [22,23]. Moreover, mutations in IDH, SDH, and FH were found to accumulate structural analogues of alpha-ketoglutarate that may inhibit the activity of prolyl hydroxylases and promote the expression of HIF target genes, and this biochemical fingerprint is also imperative to maintain cancer stem cells and to induce their differentiation [7,24,25]. Lastly, lactate is the only metabolic compound involved in all the main sequela of cancer and lactagenesis was reported to (i) inhibit lactic acid secretion from T cells and reduce their proliferation and cytokine production by 95% [26]; (ii) serve as an energy vehicle in oxidative cancer cells [6,15,26]; (iii) regulate cancer-derived exosome release, uptake, and physiology [27]; (iv) induce blood vessel invasion in response to tumour induced angiogenic factors [17,28]; (v) promote invasion and metastasis by stimulating the production of hyaluronan and the expression of its membrane receptor CD44 [29]; (vi) positively correlate with radio-resistance and reduce the cellular uptake of several anticancer drugs such as doxorubicin, mitoxantrone, and vincristine [30,31]; and (vii) sustain the conversion of glyceraldehyde-3-phosphate to 1,3-biphosphoglycerate by regenerating nicotinamide adenosine diphosphate [32].

## 3. Evolution of Small Molecules from Natural Resources

Natural compounds have been reported to have suppressive effects on the initiation, promotion, and progression of human cancers, and 60% of successful antineoplastic agents in clinical use are derived from naturally occurring compounds and their synthetic analogues [33,34]. For instance, paclitaxel, a diterpenoid isolated from *Taxus brevifolia*, is currently used in the treatment of breast, ovarian, and lung cancer, and other Food-and-Drug-Administration-approved anticancer drugs include vinblastine—a tubulin/microtubule formation inhibitor and camptothecin—a topoisomerase I inhibitor [35,36,37]. Until now, an increasing number of natural compounds have also been demonstrated to counteract aberrant metabolism in cancer cells. Kueck et al. [38] have reported that resveratrol, a stilbene extracted from grapes, induced autophagocytosis in ovarian cancer cells by blocking the phosphorylation of AKT and mTOR, while Gomez et al. [39] have highlighted that resveratrol inhibited PFK activity in breast cancer cells by promoting its dissociation to a low active dimer [33] (Figure 1). Furthermore, Vanamala et al. [40] have outlined that resveratrol also decreased the activity of glucose-6-phosphate dehydrogenase, transketolase, and phosphogluconate dehydrogenase in colorectal cancer cells. Hensley et al. [41] have reported that curcumin, a polyphenol extracted from *Curcuma longa* and being tested for the prevention and treatment of oral, head and neck, lung, and pancreas cancer, as well as osteosarcoma, glioblastoma, and chronic lymphocytic leukemia in clinical trials, decreased glucose transporter 4, hexokinase 2 (HK2), 6-phosphofructo-2-kinase/fructose-2,6-biphosphatase 3, and pyruvate kinase M2 at the mRNA and protein levels via AMPK-mediated regulation in oesophageal cancer cells [42,43] (Figure 1). Pereira et al. [44] have documented that this polyphenol also modulated the expression of transcription factors and growth factors as upstream regulators of tumour cellular bioenergetics in clinical trials. Moreira et al. [45] have highlighted that epigallocatechin gallate (EGCG), a green tea polyphenol used to prevent colorectal cancer progression in patients with curative resections, inhibited glucose uptake and the active secretion of lactic acid in a pilot trial and Wu et al. [33] have reported that the reversible inhibition of fatty acid synthase by EGCG was not accompanied by off-target effects (Figure 1). Other anticancer metabolic modulators include quercetin—a flavonoid that inhibited glycogen synthesis in pancreatic adenocarcinoma cells and berberine—an isoquinoline quaternary alkaloid that downregulated triose phosphate isomerase, aldolase A, and enolase 1 in breast cancer cells [46,47] (Figure 1). Synthetic compounds have also targeted key enzymes and transporters involved in metabolic rearrangements undertaken by cells in oncogenesis but their clinical success has been impaired by their pharmacokinetic limitations and adverse effects [7,12,48,49]. On the other hand, natural compounds have shown an ideal favourable profile, and this highlights the need to explore other bioresources and assay their bioactive metabolites to target other energy-relevant regulators in cancer.

Currently, 270 species of mushrooms are reported to be potentially useful for human health and the mushrooms credited with success against neoplasia belong to the genus *Agaricus*, *Albatrellus*, *Antrodia*, *Calvatia*, *Clitocybe*, *Cordyceps*, *Flammulina*, *Fomes*, *Funlia*, *Ganoderma*, *Inocybe*, *Inonotus*, *Lactarius*, *Laetiporus*, *Phellinus*, *Pleurotus*, *Russula*, *Schizophyllum*, *Suillus*, *Trametes*, and *Xerocomus*, and compounds isolated from these mushrooms have targeted each of the cancer hallmarks recently listed by Hanahan and Weinberg [50] in tumour cell systems and animal assays [51,52,53,54,55,56]. For example, Kang et al. [57] have demonstrated that ergosterol peroxide isolated from *Inonotus obliquus* inhibited the growth of human colorectal cancer cell lines HCT116, HT-29, SW620, and DLD-1 by suppressing the nuclear levels of beta-catenin which ultimately resulted in reduced transcription of c-MYC, cyclin D1, and cyclin-dependent kinase-8 (Figure 2). Furthermore, ergosterol peroxide administration also suppressed tumour growth in the colon of AOM/DSS-treated mice. Liu et al. [58] have observed DNA fragmentation, phosphatidylserine externalisation, caspase-3, -8, and -9 activation, mitochondrial membrane potential depolarisation, and cytochrome c release when HepG2 cells were treated with suillinthatwas derived from the mushroom *Suillusplacidus* (Figure 2). Similarly, Nakamura et al. [59] have documented that caffeic acid phenethyl ester from *Agaricus bisporus* induced the maturation of dendritic cells via nuclear factor kappa B (NF-κB), extracellular signal-regulated kinase, and p38 mitogen-activated protein kinase signalling pathways (Figure 2). Cheng et al. [60] have outlined that pachymic acid and polyporenic acid C have suppressed angiogenesis in human pancreatic adenocarcinoma cell line BxPc-3 by abrogating matrix metalloproteinase-7 synthesis (Figure 2). Lastly, Xu et al. [61] have shown that ganoderic acid T inhibited 95-D lung cancer cell migration by promoting cell aggregation and downregulating matrix metalloproteinase 2 and matrix metalloproteinase 9gene expression (Figure 2). Nevertheless, the number of myco-chemical structures under investigation for antimetabolic purposes is very poor and is at an infancy stage, and this review will offer an integrated view on the ability of these agents to target metabolic reprogramming and will also enable their potential as metabolic modulators in cancer to be critically ascertained.

## 4. Mushroom-Derived Compounds Targeting Disrupted Metabolism in Cancer

### 4.1. Neoalbaconol Induced Energy Depletion and Multiple Types of Cell Death by Targeting PDK-1-PI3K/AKT Signalling Pathway

*Albatrellusconfluens* is a member of the Albatrellaceae family and is mainly distributed in North America, Europe, and eastern Asia [52,62]. Several biologically active secondary metabolites with anticancer potential have been isolated from this polypore mushroom [63,64,65]. Recently, neoalbaconol (NA), a small-molecule with a drimane-type sesquiterpenoid structure, was extracted from the fruiting body of this fungus [66] (Figure 3).

Deng et al. [66] demonstrated that NA can significantly inhibit the proliferation of nasopharyngeal carcinoma cell lines (C666-1, HK1, SUNE1, HNE2-LMP1, CNE1-LMP1, and 5-8F), breast cancer cell lines (ZR75-1, MX-1, T47D, MAD-MB-231, MDA-MB-453, and MCF-7), colon cancer cell lines (HCT116 and SW620), leukaemia cell line (K562), prostate cancer cell line (DU145), lung adenocarcinoma epithelial cell line (A549), and melanoma cell line (A375) in a time- and dose-dependent manner. The latter did not affect the proliferation of immortalised normal cell lines (human keratinocyte HaCaT, human nasopharynx epithelial NP69, and mouse fibroblast NIH/3T3) at high doses. This underscored the selectivity of the constituent isolated from *A. confluens* toward cancer cells with C666-1, HK1, and ZR75-1 cells reported to being more sensitive to NA. Flow cytometry, protein, and confocal and electron microscope assays have shown that apoptosis and necroptosis were responsible for the death-inducing efficacy of NA and the cleavage of poly (ADP-ribose) polymerase-1 by caspases, and the interaction and colocalisation of endogenous receptor-interacting protein 1 and receptor-interacting protein 3 were observed in C666-1 cells. This is because by docking into the ATP-binding pocket of phosphoinositide-dependent kinase 1 (PDK1) and inhibiting its kinase activity, NA inhibited the PI3K/AKT/mTOR pathway and its downstream metabolic regulator HK2 and autophagy, which was activated by c-JUN N-terminal kinases to provide a survival advantage, was unable to reverse the NA-induced energy depletion stress and cell death.

Furthermore, Deng et al. [66] also evaluated the in vivo efficacy of the secondary metabolite (100 mg/kg/day) in C666-1 induced tumours in athymic nude mice. The average tumour volume was 2.4 times smaller in the NA-treated group compared with the vehicle-treated group. None of the mice showed signs of toxicity and the average tumour weight of the NA-treated group and control group was 0.65 ± 0.23 g and 1.26 ± 0.32 g, respectively, at the treatment end point. Consistent with the in vitro data, the in vivo data outlined the efficacy of NA in suppressing tumour progression by inhibiting the AKT signalling pathway (Figure 4).

### 4.2. GL22 Suppressed Tumour Growth by Altering Lipid Homeostasis and Triggering Cell Death

*Ganoderma* is a cosmopolitan genus of polypore fungi in the family Ganodermataceae and is commonly known as Ling Chu, Ling Zhi, reishi, and the mushroom of immortality [51,52,67,68,69]. These species contain polysaccharides, fatty acids, steroids, triterpenoids, and alkaloids as bioactive constituents and have been traditionally administered throughout Asia for centuries as a cancer treatment and are highly sought-after and of great economic value [70].

Wang et al. [71] isolated GL22, a triterpene farnesyl hydroquinone hybrid, from the fruiting body of *G. leucocontextum*, and Liu et al. [72] evaluated the bioactivity of this compound (Figure 3). They reported that GL22 displayed growth-inhibitory activity against Huh7.5 cells with a half-maximal inhibitory concentration (IC_50_) value of 8.9 μM and haematoxylin and eosin staining has revealed that a Huh7.5 xenograft tumour treated with GL22 displayed enlarged intercellular spaces and decreased cell density relative to the control. This natural compound exerted inhibitive effects in cell line experiments and animal modelling by altering cellular lipid homeostasis. In fact, GL22 treatment significantly decreased the expression levels of peroxisome proliferator-activated receptor alpha and peroxisome proliferator-activated receptor gamma, which subsequently led to the downregulation of the mRNA levels of fatty acid-binding protein 1, fatty acid-binding protein 4, and fatty acid-binding protein 5. These intracellular lipid chaperones reversibly bind fatty acids and coordinate their import, transport, storage, and export, and by antagonising the transcriptional levels of fatty acid binding proteins, GL22 blocked fatty acid transport. This GL22-mediated immobilisation of free fatty acids led to a sharp increase in the average number and size of lipid droplets, and this observation was consistent with impaired cardiolipin biosynthesis. This signature phospholipid of the mitochondria is involved in mitochondrial biogenesis, mitochondrial bioenergetics, and mitochondrial dynamics, and the GL22-induced abnormality in cardiolipin content altered the mitochondrial shape and size and caused membrane integrity damage and fragmentation of the mitochondrial cristae. This eventually triggered mitochondrial dysfunction, reduced ATP production, and decreased aerobic respiration in Huh7.5 cells.

Furthermore, GL22 treatment induced apoptosis in liver cancer cells by upregulating the levels of p53 and Bcl-2-associated X protein and downregulating the level of Bcl-2. Similarly, the hallmarks of the mitochondrial-mediated intrinsic apoptotic pathways, which include caspase 3, caspase 9, and poly (ADP-ribose) polymerase cleavage, and the hallmarks of the death receptor-mediated extrinsic apoptotic pathways, which include caspase 8 cleavage, were activated by GL22 treatment (Figure 5).

### 4.3. Grifolin Reversed DNMT1-Mediated Metabolic Reprogramming Induced by Epstein-Barr Virus Latent Membrane Protein 1

Epstein-Barr virus (EBV) is responsible for lymphoid and epithelial malignancies, and this oncovirus infection is characterised by the expression of latent genes, which include Epstein-Barr nuclear antigens, Epstein-Barr nuclear antigen leader protein, latent membrane protein 1 (LMP1), latent membrane protein 2, non-coding EBV-encoded RNAs, and viral microRNA [73]. Among them, LMP1 is a driver oncogene in nasopharyngeal carcinoma and plays an imperative role in its pathogenesis by upregulating HK2, activating the fibroblast growth factor 2/fibroblast growth factor receptor 1 signalling pathway, and increasing the expression of glucose transporter 1 through the mTORC1/NF-κB signalling pathway [74,75,76].

Luo et al. [77] have documented that aerobic fermentation in CNE1-LMP1 cells was markedly enhanced by 90% compared to LMP1-negative CNE1 cells, and this is primarily because LMP1 downregulated the PTEN/AKT signalling pathway in a DNA methyltransferase 1 (DNMT1)-dependent manner. Furthermore, LMP1 also switched glucose metabolism from oxidative phosphorylation to aerobic glycolysis in nasopharyngeal carcinoma cells by promoting DNMT1 mitochondrial translocation, which further led to an increase in the methylation/unmethylation (M/U) ratio of DNA fragments in the mitochondrial DNA D-loop region and a decrease in the DNA levels of MT-COXII, MT-ATP6, and MT-ND6 that encode cytochrome c oxidase subunit II, ATP synthase F_0_ subunit 6, and NADH dehydrogenase 6, respectively. These observations were consistent with the metabolic flux measurements, which showed that the ^13^C-labelled lactate level was significantly increased and the ^13^C-labelled TCA cycle metabolite levels, which included ^13^C-α-ketoglutarate, ^13^C-citrate, ^13^C-fumarate, and ^13^C-malate, were significantly decreased in CNE1-LMP1 cells compared with CNE1 cells.

Grifolin, a farnesyl phenolic compound extracted from the mushrooms *A. confluens* and *Boletus pseudocalopus*, which previously suppressed the growth and metastasis of HeLa, MCF-7, SW480, K562, and MG63 tumour cell lines, was reported to decrease the amount of glucose used for lactate production in LMP1-positive cells by targeting DNMT1 to demethylate and reactivate the PTEN gene (Figure 3). Furthermore, grifolin treatment significantly attenuated DNMT1 retention in the mitochondria of CNE1-LMP1 cells, and the M/U ratio of the mitochondrial DNA D-loop region was consequently decreased in these cells. Similarly, grifolin promoted subunit assembly to form oxidative phosphorylation complexes, and it also ensured that more nicotinamide adenine dinucleotide is used by complex I to enhance respiration in CNE1-LMP1 cells (Figure 6).

Luo et al. [77] also reported that grifolin can phenocopy the effect of DNMT1 inhibitor 5-AZA-2-deoxycytidine and, unlike this epidrug, grifolin is chemically stable and not accompanied by serious side effects and, thus, can be used as a safe alternative to improve tumour control.

### 4.4. Conjugated Linoleic Acid Exhibited Proapoptotic Effects by counteracting Altered Lipid Metabolism

*Agaricus* is the type genus of the family Agaricaceae in the phylum Basidiomycota and is distributed worldwide [78,79,80]. These saprobic mushrooms are often gregarious in forests, pasture land, grass land, rubbish dumps, manure heaps, and alluvial soil and include economically important species, such as *A. bisporus*, which is also known as white button mushroom, table mushroom, cultivated mushroom, portobello mushroom, and crimini mushroom [78,81,82,83,84]. This genus of macrofungi is widely used and extensively studied for its dietetic, ethnopharmacological, and medicinal properties, and the literature abounds on the topic [85,86,87,88,89,90].

Chen et al. [91] have identified conjugated linoleic acid (CLA) as the main constituent of an ethyl acetate extract of *A. bisporus*,and they reported that CLA decreased breast cancer cell proliferation by inhibiting aromatase activity(Figure 3). Adams et al. [92] investigated the anticancer potential of CLA in androgen-sensitive LNCaP and androgen-insensitive PC3 and DU145 prostate cancer cell lines. The mushroom extract decreased cell proliferation in all cell lines tested in a dose-dependent manner compared with untreated control cells, and the magnitude of response was similar between the cell lines. The Cell Death Detection Elisa^Plus^ Photometric Enzyme Immunoassay and the Annexin V Assay results highlighted that the mushroom extract induced DNA fragmentation and phosphatidylserine translocation in the prostate cancer cell lines, but LNCaP cells were more prone to the proapoptotic effects of CLA, and this underscored that the antiproliferative activity is different from the proapoptotic activity of the extract because the cell lines under study responded equally in the Real-Time Proliferation Assay. Furthermore, Adams et al. [92] also evaluated the prostate cancer protective effects of *A. bisporus* in vivo by using male athymic mice injected with either PC3 or DU145 prostate cancer cells, andthe LNCaP cell line was not used for animal experiments because it did not form tumours when implanted. The in vivo studies have shown that the oral intake of the mushroom extract decreased PC3 tumour weight and DU145 tumour weight by 68.6% and 44.5%, respectively, compared with the pair-fed control mice. Histological examinations of PC3 and DU145 tumours have revealed that cell proliferation decreased by 45% and 25.3%, respectively, in the mushroom-extract-fed group compared with the control, and the level of apoptosis was significantly increased in DU145 tumours.

The in vitro results were in line with the in vivo results, and the microarray analysis suggested several mechanisms for the CLA effect on prostate cancer cell proliferation and apoptosis. The CLA-rich mushroom extract upregulated the expression of FAS/APO-1 gene which plays a central role in the physiological regulation of apoptosis by 2.84-fold and downregulated the expression of KIT gene, which is involved in the proliferation and survival of cells by four-fold. Similarly, the extract inhibited diacylglycerol production and eventually increased apoptosis through increased production of arachidonic acid and ceramide and also decreased cyclooxygenase-2 protein expression and subsequently suppressed the conversion of arachidonic acid to prostaglandin E2, which fosters cancer progression by (1) blocking apoptosis through the activation of the PI3K/AKT/PPAR signalling pathway, (2) increasing angiogenesis through vascular endothelial growth factor activation, (3) inducing immune suppression by increasing interleukin 10 production, and (4) increasing cancer cell proliferation through the activation of rat sarcoma virus/rapidly accelerated fibrosarcoma/mitogen-activated protein kinase kinase/extracellular signal-regulated kinase signalling pathway. The aforesaid effects on lipid metabolism were accompanied by the inhibition of isocitrate dehydrogenase 2 (IDH2) and the increased expression of FH. The three-fold decrease in IDH2 expression normalised the TCA cycle in prostate cancer cells, and the 4.3-fold increase in FH expression inhibited fumarate build-up in prostate cancer cells and deactivated the angiogenic factor hypoxia-inducible factor 1-alpha (Figure 7).

Adams et al. [92] also observed that the ethyl acetate extract downregulated the expression of the endothelin 1 gene, which is commonly upregulated in hypoxia by three-fold. Furthermore, the extract also upregulated the expression of immune-related genes, such as those encoding interleukin 15, S100 calcium binding protein A8, S100 calcium binding protein A9, and lectin, galactosidase-binding soluble, 1 proteins.

### 4.5. (22E, 24R)-6β-methoxyergosta-7,9(11),22-triene-3β,5α-diol as a Potential Non-Competitive Inhibitor of HK2

Bao et al. [93] identified two new steroids: tetraoxycitricolic acid (1) and (22E, 24R)-6β-methoxyergosta-7,9(11),22-triene-3β,5α-diol (2) from a solvent extract of *G. sinense*(Figure 3). The structure-based virtual ligand screening has shown that the steroid 2 had the highest binding affinity to HK2, and the microscale thermophoresis technique revealed that steroid 2 displayed an equilibrium dissociation constant of 114.5 ± 2.7 μM, which has added further credibility to this observation. Furthermore, the molecular docking result highlighted that steroid 2 occupied half of the product releasing pathway, and two hydrogen bonds were predicted between threonine 536 and 3-hydroxyl group and arginine 539 and 5-hydroxyl group, respectively, and key hydrophobic interactions between methionine 555 and the terminal isopropyl group were also observed. Similarly, in vitro studies have shown that steroid 2 exhibited inhibitory effects against HK2 with an IC_50_ value of 2.06 ± 0.15 μM via a non-competitive inhibition. Lastly, steroid 2 exhibited four-fold selectivity against human pancreatic ductal adenocarcinoma (PDAC) SW1990 cells and normal African green monkey kidney cells, and the IC_50_ value of steroid 2 against the PDAC cell line was 5.05 ± 0.17 μM, which was less than that of known inhibitors, such as metformin and benserazide.

### 4.6. Panepoxydone Exerted Antiproliferative Effects by Downregulating LDHA Expression and Reinducing LDHB Expression

Arora et al. [94] have shown the IC_50_ values of panepoxydone (PP), a natural NF-κB inhibitor isolated from the edible mushroom *Lentinus crinitus*, as 4 μM in MCF-7 breast cancer cell line and 5 μM, 8 μM, and 15 μM in triple negative breast cancer (TNBC) cell lines MDA-MB-453, MDA-MB-468, and MDA-MB-231, respectively (Figure 3). Furthermore, they have argued that the PP treatment reduced the viability of the breast cancer cell lines by targeting deregulated energy processing because the cell metabolism assays revealed that MCF-7, MDA-MB-453, MDA-MB-468, and MDA-MB-231 cells showed decreases in oxygen consumption rate (OCR), and MDA-MB-453 and MDA-MB-231 cells showed decreases in extracellular acidification rate (ECAR) after 24 h of exposure to the fungal secondary metabolite. The OCR to ECAR ratio, which is an indicator of oxidative phosphorylation, was also increased in TNBC cells. The ATP-based bioluminescence assay also confirmed the PP-mediated energy depletion stress in MCF-7 and TNBC cells.

In fact, the PP-induced alterations in the breast cancer cells bioenergetics fostered protective mechanisms against cancer initiation and progression by abrogating lactate dehydrogenase A (LDHA) expression levels and inducing the re-expression of lactate dehydrogenase B (LDHB) through demethylation of the enzyme promoter to similar levels observed in normal human mammary epithelial cells, which further led to the accumulation of pyruvate, the activation of the respiratory chain, the generation of more electrons from the chain, and enhanced levels of ROS which then decreased the mitochondrial membrane potential, damaged the mitochondrial membrane, and triggered the mitochondrial pathway of apoptosis. Similarly, LDHB overexpression also reduced cell migration in MCF-7 and TNBC cells by 2–3-fold relative to empty vector cells (Figure 8).

It is important to note that Arora et al. [94] highlighted that the non-productive mitochondrial respiration and mitochondrial pathway-induced apoptosis caused by PP are in conjunction with studies conducted with FX-11, 3-BrPA, and 2-DG.

## 5. Conclusions

The remodelling of cancer metabolism and bioenergetics is a universal alteration in oncogenesis and these metabolic enzymes are interesting targets of cancer therapy. The ongoing search for energy-relevant regulators in cancer led to metabolites isolated from mushrooms. Unlike the synthetic metabolic inhibitors, these mushroom-derived compounds have demonstrated excellent differential toxicity and pharmacological properties. They have also modulated a wider range of aberrantly regulated metabolic keynodes in cancer and were needed in relatively lower doses to exhibit therapeutic effects. This ideal favourable profile supports the feasibility of these molecules in anticancer therapies and chemopreventive strategies. Hence, a future area of research should focus on long-term, double-blind, and placebo-controlled clinical trials to delineate their possible roles in the prevention and management of cancer. There is also a need to explore scalable synthetic routes for the production and supply of these compounds especially if clinical trials are successful.

## Figures and Tables

**Figure 1 molecules-28-01441-f001:**
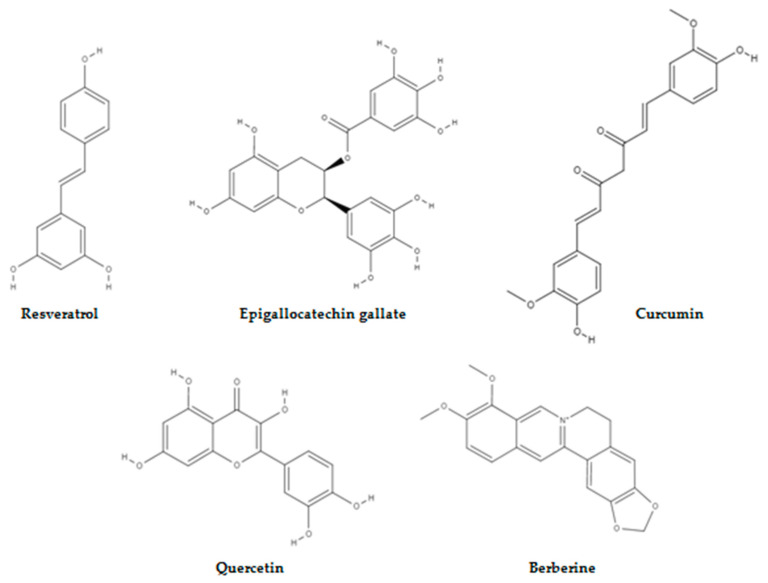
Naturally occurring compounds that target aberrant metabolism in cancer.

**Figure 2 molecules-28-01441-f002:**
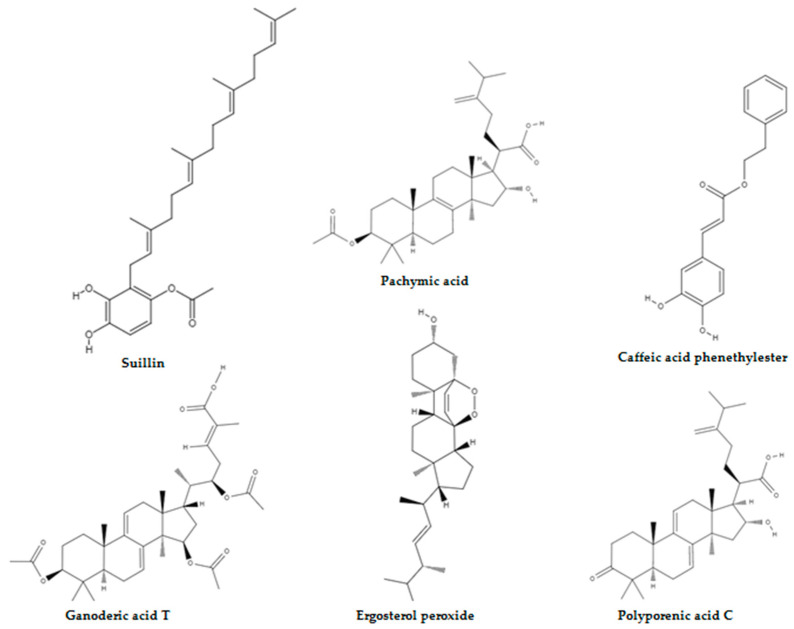
Mushroom-derived compounds that have anticancer potential.

**Figure 3 molecules-28-01441-f003:**
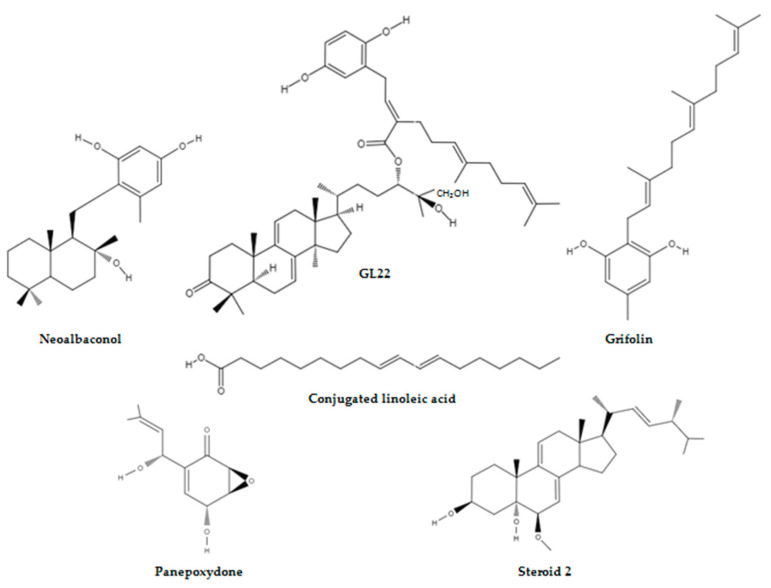
Structures of mushroom-derived compounds that target energy-relevant regulators in cancer.

**Figure 4 molecules-28-01441-f004:**
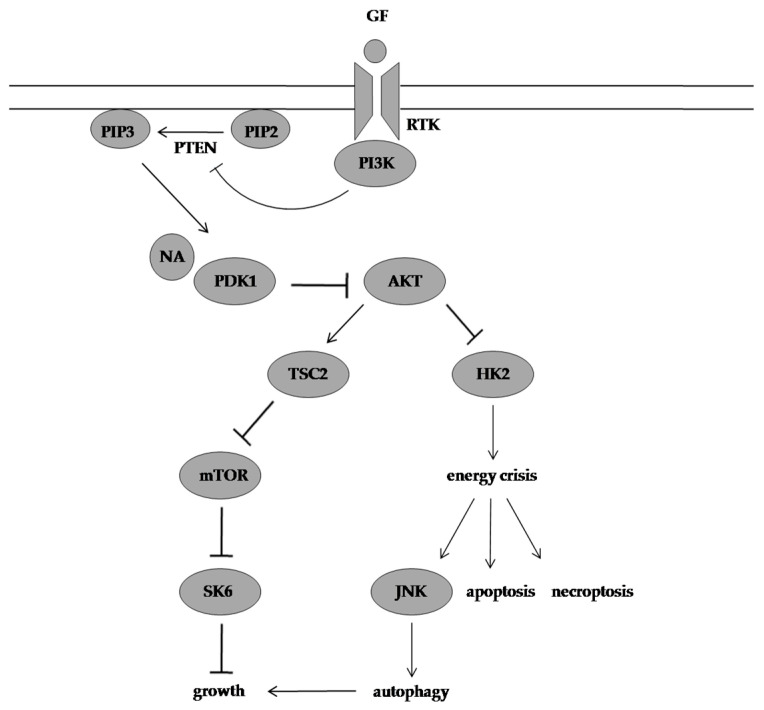
Molecular mechanism of action of NA. NA targets PDK1 and suppresses its downstream effectors resulting in energy crisis, and autophagy provides a survival advantage in NA-treated cells but is unable to prevent cell death in response to this energy depletion. GF:growth factor, RTK: receptor tyrosine kinase, PI3K: phosphoinositide 3-kinase, PIP2: phosphatidylinositol-4,5-biphosphate, PTEN: phosphatase and tensin homolog, PIP3: phosphatidylinositol-3,4,5-triphosphate, NA:neoalbaconol, PDK1: phosphoinositide-dependent kinase 1, AKT:protein kinase B, TSC2: tuberous sclerosis 2, mTOR: mammalian target of rapamycin, SK6: p70 S6 kinase, HK2: hexokinase 2, JNK: c-JUN N-terminal kinase.

**Figure 5 molecules-28-01441-f005:**
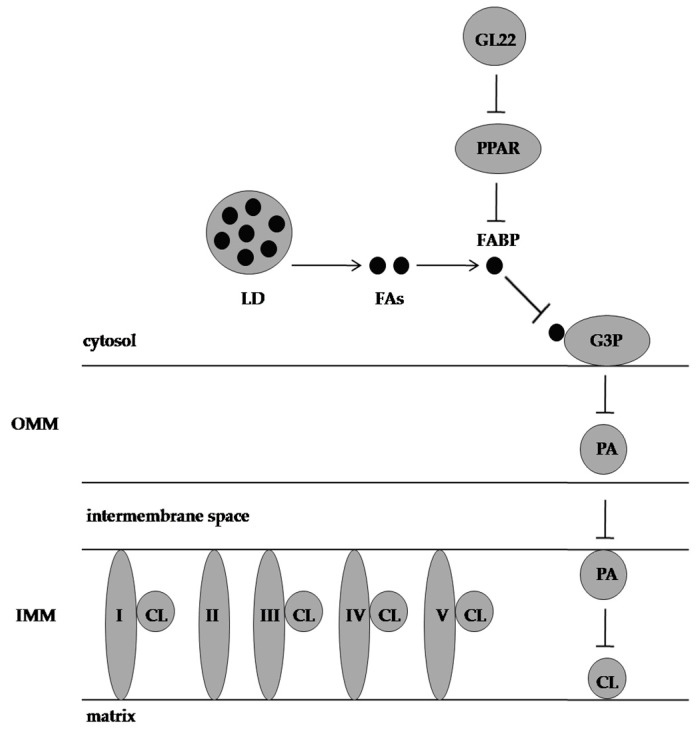
Effect of GL22 on Huh7.5 cells. CL is needed for oxidative phosphorylation complexes assembly and is synthesised from G3P and FAs. GL22 displays antimetabolic activity by downregulating FABP expression and inhibiting CL biosynthesis. PPAR: peroxisome proliferator-activated receptor, LD: lipid droplet, FAs: fatty acids, FABP: fatty-acid-binding protein, G3P: glycerol-3-phosphate, PA: phosphatidic acid, CL: cardiolipin, OMM: outer mitochondrial membrane, IMM: inner mitochondrial membrane, I, II, III, IV, and V represent electron transport chain complexes.

**Figure 6 molecules-28-01441-f006:**
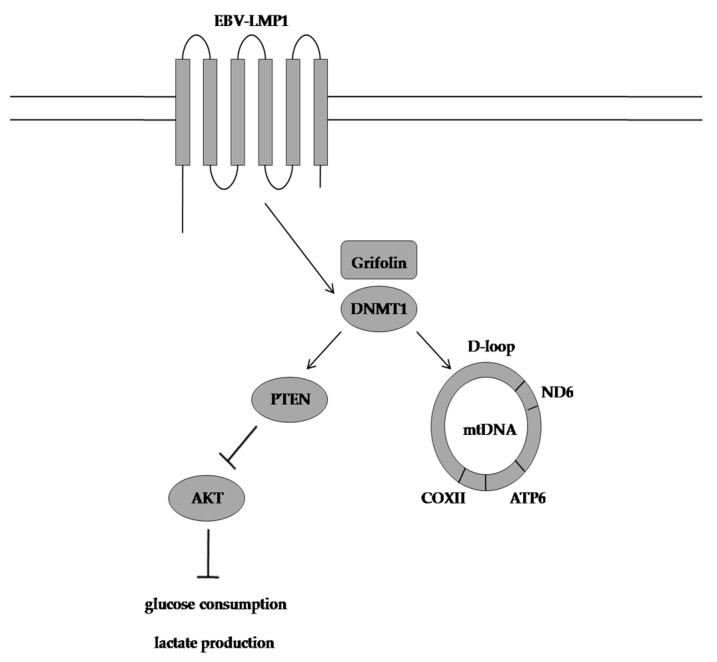
Grifolin-induced mode of action. Grifolin restores oxidative phosphorylation in nasopharyngeal carcinoma cells by reactivating the PTEN gene and reducing the M/U ratio of the mitochondrial DNA D-loop region. EBV-LMP1: Epstein-Barr virus latent membrane protein 1, DNMT1:DNA methyltransferase 1, PTEN: phosphatase and tensin homolog, AKT:protein kinase B, COXII cytochrome oxidase subunit II, ATP6:ATP synthase membrane subunit 6, ND6:NADH ubiquinone oxidoreductase core subunit 6, M/U ratio:methylation and unmethylation ratio.

**Figure 7 molecules-28-01441-f007:**
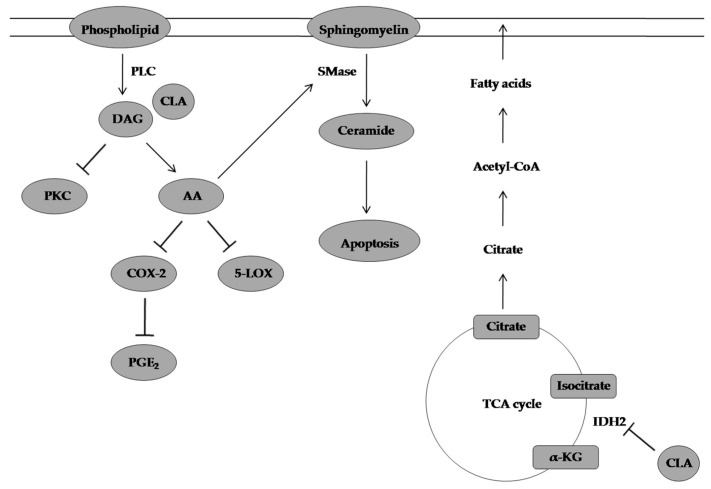
CLA-inducing apoptosis in prostate cancer cell lines by inhibiting DAG production, increasing AA and ceramide production, and inhibiting IDH2 activity. PLC: phospholipase C, CLA: conjugated linoleic acid, DAG: diacylglycerol, PKC: protein kinase C, AA: arachidonic acid, COX-2: cyclooxygenase-2, 5-LOX: 5-lipooxygenase, PGE2: prostaglandin E2, SMase:sphingomyelinase, α-KG: alpha-ketoglutarate, IDH2: isocitrate dehydrogenase.

**Figure 8 molecules-28-01441-f008:**
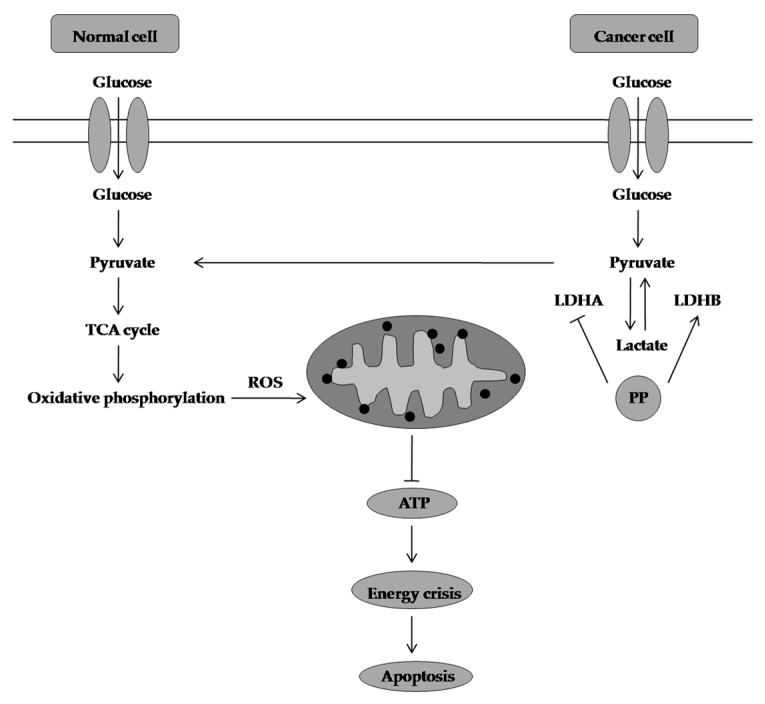
Action of PP in breast cancer. PP downregulates LDHA expression and upregulates LDHB expression. The ROS generated during oxidative phosphorylation decreases the ∆ψm and this leads to energy crisis and cell death. TCA cycle: tricarboxylic acid cycle, ROS: reactive oxygen species, ATP: adenosine triphosphate, PP:panepoxydone, LDHA: lactate dehydrogenase A, LDHB: lactate dehydrogenase B, ∆ψm: mitochondrial membrane potential.

## Data Availability

Not applicable.

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
