# Peer review of "Mushroom-Derived Compounds as Metabolic Modulators in Cancer"

_molecules, 2023, doi:10.3390/molecules28031441_

Round 1

Reviewer 1 Report

The title of the manuscript is catchy and tends to interest the reader, but the content of the manuscript is not.

Key issues.

1. the abstract section is general and fails to focus and give effective information.

2. 3.2 and 3.3 of the third part (3. Evolution of small molecules as metabolic modulators in cancer) overlap in scope. Isn't "Mushroom-derived compounds" part of "Naturally occurring compounds"? The keyword also has this problem. 3.

3. There are numerous natural products with anticancer properties and proven by in-depth pharmacological studies, and there are many such compounds from mushroom sources alone. As a review article, the comprehensiveness of information is a basic requirement.

4. The fourth part (4. Mushroom-derived compounds targeting disrupted metabolism in cancer) is better organized, summarizing the anti-cancer mechanisms of related compounds according to the types of signaling pathways. However, the problem is that these compounds are Mushroom-derived compounds, and the authors should review them carefully.

5. There is a big problem in the writing of the manuscript, the logic of the second part is confusing. Synthetic compounds" are inappropriate.

All in all, the manuscript has an eye-catching title, but unfortunately, the content of the manuscript does not support such a title. It is suggested that the authors focus on Mushroom-derived compounds with anticancer properties and their mechanisms, as in the current title. Focus on this point and rewrite the manuscript.

Reviewer 2 Report

The manuscript “Mushroom-derived compounds as metabolic modulators in cancer” by Dowaraka-Persad and Neergheen presents several examples of the use of mushroom-derived compounds as metabolic modulators in cancer. Due to the relevance of the modulation of metabolic pathways in the development of possible treatments for cancer, the manuscript could be of relevance for researchers on this field.

However, the present review manuscript has some important issues that must be addressed before it can be considered for publication.

Firstly, the information is not organized correctly. The manuscript must be rewritten and rearranged so that the information is clearly displayed.

Mushroom derivatives are proposed as modulators of metabolism in cancer cells. It is OK to start in section 2 by introducing the metabolism of cancer cells and in section 3 to talk about the small molecules that have been used as modulators. Section 3.1 is correct, synthetic compounds. The 3.2 would be natural compounds. Why Naturally occurring compounds and metabolic targets?

If section 3 is metabolic modulators in cancer, 3.3 would be mushroom compounds (as metabolic modulators in cancer). But the section 4 is Mushroom-derived compounds targeting disrupted metabolism in cancer… As mushroom compounds are the main aim of the review, authors must introduce mushroom compounds and their importance in pharmacology, as antitumour and then as metabolic modulators.

Other issues:

Page 3, line 99: the first time a compound is named you should state in which figure you can find the structure.

Page 3, line 110: change 2-deoxyglucose to 2-Deoxyglucose

Figure 1: All drawings must use the same settings; in this figure, different line settings and atom sizes

Page 4, line 138: not all natural compounds are inexpensive. Low concentrations/extraction procedures can result in rather expensive isolated materials!

Round 2

Reviewer 1 Report

After revision by the authors, the quality of the manuscript has improved considerably and it can be considered for publication in this journal. The following issues, including but not limited to, still need to be revised.

1.       What does steroid 2 refer to? Please indicate where appropriate.

2.       The chemical structures in Figures 1 to 3 are still so ugly, please refer to the submission guidelines for redrawing.

3.       For the description in lines 145-151, please add citations to the following literature to strengthen the supporting evidence.

a.         Diverse Metabolites and Pharmacological Effects from the Basidiomycetes Inonotus hispidus (vol 11, 1097, 2022), ANTIBIOTICS-BASEL, 0.3390/antibiotics11111671

b.        Genomic and Metabolomic Analyses of the Medicinal Fungus Inonotus hispidus for Its Metabolite's Biosynthesis and Medicinal Application, JOURNAL OF FUNGI, 10.3390/jof8121245

c.         Chromosome-Level Genome Sequences, Comparative Genomic Analyses, and Secondary-Metabolite Biosynthesis Evaluation of the Medicinal Edible Mushroom Laetiporus sulphureus, Microbiology Spectrum, 10.1128/spectrum.02439-22

Author Response

The authors are grateful for the comments and suggestions from the reviewer.

  1. What does steroid 2 refer to? Please indicate where appropriate.

Steroid 2 has been replaced by its chemical name in the revised manuscript. Please see Page 1 Line 32 and Page 11 Line 382.  

  1. The chemical structures in Figures 1 to 3 are still so ugly, please refer to the submission guidelines for redrawing.

The chemical structures have been drawn using the software MolView and we have checked the consistency of the structures. 

  1. For the description in lines 145-151, please add citations to the following literature to strengthen the supporting evidence.

The citations have been included in the revised manuscript.

Reviewer 2 Report

I believe authors did a really good job with the revision, the manuscript has been greatly improved.

In my opinion, the manuscript is ready for publication 

Author Response

We thank the reviewer for accepting the manuscript for publication.